# RegCLIP: A Label-Efficient Coarse-to-Fine Learner for Ordinal Regression

## Abstract

Ordinal regression is a fundamental problem within the field of computer vision. While pre-trained vision-language models have exhibited impressive performance on various vision tasks, their potential for ordinal regression has received less exploration. In this paper, we introduce a novel method called RegCLIP, a label-efficient coarse-to-fine method for ordinal regression. This approach incorporates language prior information to gradually refine predictions and achieve fine-grained results. Our RegCLIP framework encompasses two levels of coarse-to-fine concepts. The first level is a stagewise approach, performing intermediate classification initially and then refining the predictions. The second level is to generate coarse semantic labels as intermediate classes and subsequently refine them into fine-grained labels. To achieve it, we propose a ***novel coarse semantic label generation via large language models***, which generates coarse labels. To further enhance the precision of predictions, we propose a ***novel fine-grained cross-modal ranking-based loss*** specifically designed to update fine-grained semantic labels with both semantic and ordinal alignment. Experimental results on three general ordinal regression tasks demonstrate the effectiveness of RegCLIP, exceeding state-of-the-art methods with a large margin, with 10% overall accuracy improvement on historical image dating, 1.74% overall accuracy improvement on image aesthetics assessment, and 1.33 MAE reduction on age estimation under 1-shot setting.

## 1 Introduction

Ordinal regression, also known as ordinal classification, is a machine learning task designed to predict labels that naturally have an ordinal order. For example, age estimation involves predicting the age of a facial image, which follows a natural order of progression (Niu et al., 2016). Similarly, historical image dating aims to predict the decade of a given historical colored image, which also has an inherent ordinal structure (Palermo et al., 2012). Ordinal regression is a fundamental problem and has received increasing attention (Li et al., 2021; Deng et al., 2021; Lee et al., 2022).

Current techniques for ordinal regression can be categorized into three main groups: regression-based, classification-based, and ranking-based methods. Regression-based methods involve the direct estimation of a scalar value, typically achieved by minimizing the difference through loss functions such as Euclidean loss (e.g., $\ell_1$ or $\ell_2$ loss) (Yi et al., 2014). Although these methods are straightforward, they often suffer from suboptimal performance compared to classification-based methods (Rothe et al., 2018). In early classification-based approaches, cross-entropy loss is commonly used to optimize the network, treating the different categories as independent classes (Rothe et al., 2015). Recent works have taken into account the ordering relationship among labels. This is achieved through reformulating the single one-hot label as a label distribution (Gao et al., 2017; Pan et al., 2018) or re-weighting labels based on the continuous relationship (Li et al., 2019). Ranking-based methods, despite receiving comparatively less attention than the other two categories, provide an alternative approach. The underlying concept behind ranking-based methods is that making relative comparisons between samples and anchors should be more manageable than making direct predictions. Therefore, order learning algorithms can be employed in such tasks (Lee & Kim, 2022; Shin et al., 2022).

The methods mentioned above primarily focus on learning ordinal concepts within the image domain. These models are often pre-trained on datasets like ImageNet or task-specific datasets, such as IMDB-WIKI for age estimation (Rothe et al., 2018). This emphasizes the significance of using pre-trained

models for ordinal regression. Recent advancements in pre-trained vision-language models (VLMs), exemplified by CLIP (Radford et al., 2021), have shown significant improvements across various downstream tasks, including image classification (Zhou et al., 2022), semantic segmentation (Zhou et al., 2023), object detection (Gu et al., 2021), video understanding (Lin et al., 2022), etc. Nonetheless, the potential of CLIP in ordinal regression, a fundamental yet crucial task in computer vision applications, remains largely underexplored.

A straightforward approach to applying CLIP for ordinal regression is to treat the numerical index label as the class token and utilize zero/few-shot learning techniques to make predictions. Despite the promising results achieved in CLIP-based common classification or segmentation tasks, which heavily rely on the model's recognition ability, its performance on ordinal regression is notably limited. For instance, in age estimation, the zero-shot mean absolute error (MAE) is 6.09, and for historical image dating, the zero-shot accuracy is merely 26.08%; see results of "Zero-shot CLIP" (Radford et al., 2021) in Table1 and Table 2. While general prompt learning techniques like CoOp (Zhou et al., 2022) aim to enhance model adaptation to downstream tasks through trainable prompt embeddings, there still exists a significant performance gap when compared to state-of-the-art customized ordinal regression methods such as POE (Li et al., 2021) and MWR (Shin et al., 2022). Recently, OrdinalCLIP was proposed by Li et al. (2022) for the task of ordinal regression. Its main contribution is the incorporation of the ordering relationship of labels during the prompt construction stage, achieved by linearly initializing a set of learnable prompts. While this method shows promising improvements, it still exhibits two main limitations. Firstly, the ordinal constraint solely applied to the prompt side is relatively weak in guiding the consistent alignment of cross-modal features in terms of their ordinal relationships. Secondly, learning the prompt from scratch may not fully exploit the inherent knowledge embedded within the pre-trained model. In this study, we attribute the unsatisfactory performance of CLIP-based ordinal regression to two main reasons: ***the presence of insufficient numerical captions and the utilization of an ineffective training objective***.

To address these challenges, we propose RegCLIP, a novel approach for label-efficient coarse-to-fine training of CLIP for ordinal regression. Our coarse-to-fine method encompasses two key aspects: ***The first aspect of coarse-to-fine is a stagewise training approach.*** We recognize that learning from a staged process is often more effective than directly learning from precise values. Therefore, in the coarse stage, we perform intermediate classification using a limited set of classes. This allows for initial decision-making at a coarse level. Subsequently, the fine stage refines the decision within the coarse group assigned by the previous stage. ***The second aspect of our coarse-to-fine approach is to generate coarse semantic prompts and subsequently refine them into fine-grained prompts.*** To achieve this, we introduce a novel method called ***coarse semantic label generation via large language models (LLMs)*** for generating reasonable intermediate descriptions as class prompts, serving as intermediate labels during the training process. Meanwhile, this process mitigates the insufficient numerical captions issue via label transformation. To further improve the accuracy of predictions towards the fine-grained ground truth, we introduce a ***novel fine-grained cross-modal ranking-based feature regularization loss***. This loss function takes into account the inherent ordinal nature of regression labels and aims to encourage both semantic and ordinal alignment within CLIP's feature space. Its purpose is to refine the coarse semantic labels, resulting in more precise and fine-grained predictions. Overall, our RegCLIP method combines the advantages of coarse-to-fine training, coarse semantic label generation via LLMs, and fine-grained feature regularization to improve the performance of CLIP for ordinal regression. Detailed experiments show that RegCLIP outperforms the prior arts on three widely used benchmarks, with 10% overall accuracy improvement on historical image dating and 1.74% overall accuracy improvement on image aesthetics assessment, and demonstrates a clear improvement on age estimation even in a few-shot data regime. Our code will be released after paper acceptance.

## 2 RELATED WORK

**Ordinal Regression.** The goal of ordinal regression is to learn a rule to map an input image to a rank on an ordinal scale. Regression-based approaches typically employ a Euclidean loss to estimate the precise value, penalizing the disparity between the predicted values and ground-truth labels. For instance, in the work of Yi et al. (2014), a multi-scale network is proposed to directly estimate ages using an $\ell_2$ loss. However, these methods often yield subpar performance as they fail to account for the ordinal relationship among labels. Classification-based approaches partition the numbers

into distinct groups and subsequently treat the estimation of group labels as independent classes. In the work of Rothe et al. (2015), age estimation is formulated as a deep classification task, where the results are further refined using *softmax*-normalized probabilities. This strategy has been shown to outperform direct regression methods, leading to improved performance. The aforementioned studies fail to consider the inherent ordering relationship among labels. In contrast, Gao et al. (2017) address this limitation by modeling the label distribution as a normal distribution centered around the true value and subsequently perform multi-class classification. Ranking-based approaches, on the other hand, treat the original labels as rank-ordered data and compare the input with multiple reference instances. For instance, in the work of Shin et al. (2022), a moving window regression algorithm is proposed. This algorithm constructs a search window comprising two reference instances and iteratively estimates the relative rank of the input image within the window. Unlike previous works that are solely pre-trained on ImageNet or task-specific datasets within the image domain, our method, RegCLIP, leverages the rich cross-modal image-text knowledge to enhance ordinal regression performance.

**CLIP in Regression.** Recently, there are several attempts to employ CLIP for various regression topics, including depth estimation (Zhang et al., 2022), crowd/object counting (Liang et al., 2023; Paiss et al., 2023), and ordinal regression (Li et al., 2022). Depth estimation is a dense prediction task to infer the depth for each pixel, and normally the physical environment variance, depth changing rate with distance, and strategies for minimizing computational complexity are considered to ensure the satisfactory performance (Fu et al., 2018; Bhat et al., 2021). Similar situations exist for counting tasks, such as considering the object density variation in different areas (Liu et al., 2019a). DepthCLIP (Zhang et al., 2022) investigates CLIP-based monocular depth estimation in a zero-shot manner, while CrowdCLIP (Liang et al., 2023) explores CLIP's potential on crowd counting. CountingCLIP (Paiss et al., 2023) proposes a counting-contrastive loss to teach CLIP to count to ten. In general, these two tasks are well-defined domain-specific tasks. In contrast, ordinal regression is a fundamental task that involves estimating category labels with an inherent ordering relationship. This task is applicable to various domains, such as age estimation, historical image dating, and image aesthetics assessment. In this study, our focus is specifically on the task of ordinal regression. OrdinalCLIP (Li et al., 2022) tackles the ordinal regression problem by constructing a pair of learnable word embeddings that possess numerical continuity to preserve the ordinal property of text embeddings. However, it overlooks the hidden numerical knowledge and ordinal property inherent in the cross-modal feature space. In contrast, our approach, RegCLIP, focuses specifically on CLIP-based ordinal regression and introduces the concept of coarse-to-fine learning with label efficiency.

**Compositionality of VLMs.** Despite the impressive achievements, more recent works point out that such VLMs like CLIP have a weak understanding of fine-grained concepts like relational, compositional, and contextual reasoning (Radford et al., 2021; Paiss et al., 2023; Kamath et al., 2023; Xu et al., 2023; Paiss et al., 2022). Radford et al. (2021) state that CLIP is poor on fine-grained classification tasks and struggles with more systematic and abstract concepts such as counting the exact number of objects from an image. Paiss et al. (2022; 2023) demonstrate that CLIP only partially captures the meaning of input text, which attends to a limited set of its input, mainly the nouns, and is less responsive to prepositions, numbers, and adjectives. Kamath et al. (2023) find that the text encoder of CLIP falls short on attribute-object association, negation, object relationship and counting. Yuksekgonul et al. (2022) find that current VLMs have a poor relationship understanding, blunder when linking objects with attributes, and demonstrate a severe lack of order sensitivity. Such models fail to perform beyond the chance level at simple tasks requiring compositional understanding. Xu et al. (2023) focus on granularity and correctness of zero-shot recognition ability of VLMs and conclude that there is still a long way to use VLMs for zero-shot visual recognition in the real open-world setting. While recent works have primarily focused on addressing compositional limitations, effective improvements for CLIP-based ordinal regression have received less attention.

## 3    METHOD

### 3.1    PROBLEM STATEMENT

The goal of ordinal regression is to learn a rule to predict labels from an ordinal scale. Instead of directly applying a Euclidean loss for regression, the popular baseline is to treat the ordinal regression

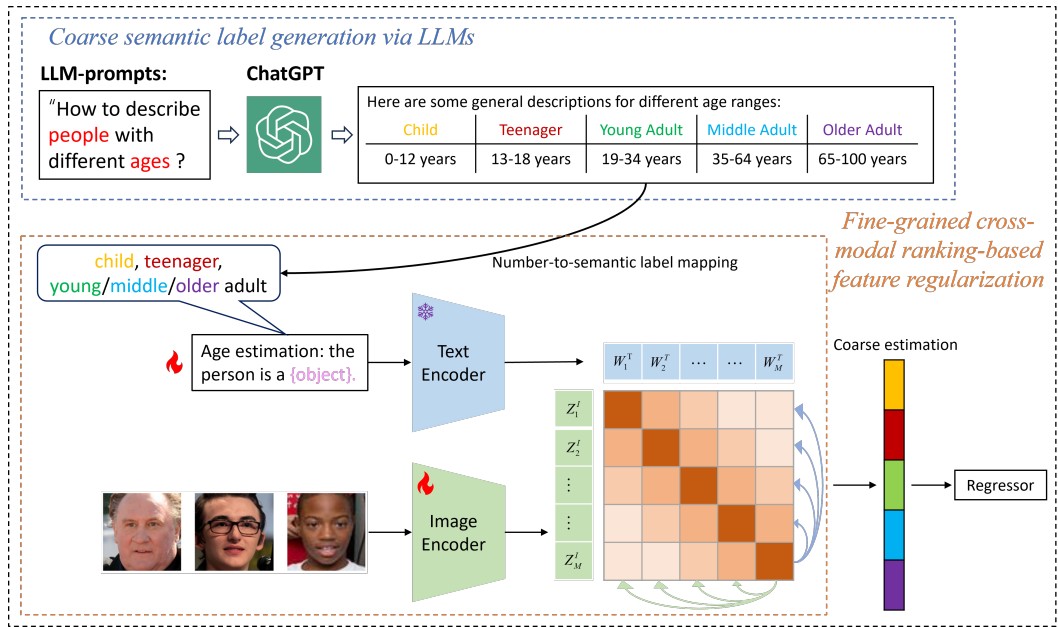

Figure 1: The framework of RegCLIP, a label-efficient coarse-to-fine learner for ordinal regression. The coarse-to-fine method encompasses two key aspects: one is a stagewise training approach by first performing intermediate classification and then refining the final predictions; the other is to first generate coarse semantic labels via LLMs, and subsequently update them via end-to-end training. A novel fine-grained cross-modal ranking-based feature regularization loss is designed to ensure both semantic and ordinal alignment in CLIP's feature space.

problem as a classification task by discretizing the labels as different bins and treating each bin as an independent class. After problem reformulation, typical multi-class classification losses like cross-entropy loss are adopted for model training and the final prediction values could be obtained by choosing the class index with highest probability or linearly multiplying all class probabilities with corresponding index values. Extra regularization might be considered by utilizing the ordinal relationship among labels. Mathematically, let $x_i$ denote the $i$ input instance with corresponding discretized label $y_i$, ordinal regression aims to recover $y_i$ by encoding the input image into feature $z_i = \Phi(x_i)$ and using a classifier $f(\cdot)$ to compute class probability $p_i$. The model is optimized with cross-entropy loss. The adaptation of VLMs like CLIP to boost ordinal regression could be achieved via image-text matching in a coarse-to-fine manner. Specifically, image feature $z_i$ could be extracted using a pre-trained CLIP image encoder $Image(\cdot)$ by $z_i = Image(\cdot)$. For a given ordinal regression task, task-related text description template $R_i$ is constructed based on linguistic mapping via LLMs. Such templates are converted into fixed-size tokens with $Tokenizer(\cdot)$ and mapped into text embeddings $w_i$ with CLIP text encoder $Text(\cdot)$. The process can be formulated as: $w_i = Text(Tokenizer(R_i))$. These text embeddings are regarded as classifier weights in a typical classification task to make initial prediction. After that, a lightweight MLP regressor $g(\cdot)$ is concatenated to refine the coarse estimation into fine-grained predictions.

## 3.2 Coarse-to-Fine Ordinal Regression Paradigm

In general, we design a coarse-to-fine CLIP-based ordinal regression paradigm. The motivation for this is based on the fact that learning from a staged process is often more effective and easier than directly learning from multiple precise values. Therefore, in the coarse stage, we perform intermediate classification using a limited set of class semantics that are generated by LLMs and consistent with ground-truth labels, which allows for initial decision-making at a coarse level. Subsequently, the fine stage refines the decision within the class group assigned by the previous stage. The whole pipeline is a stagewise approach that can be trained end-to-end, as shown in Figure 1.

## 3.3 Coarse Semantic Label Generation via LLMs

A straightforward way to leverage VLMs for ordinal regression task is to treat each rank number as an independent class token and conduct zero/few-shot learning like other downstream tasks. Unfortunately, despite the remarkable success of CLIP-based image classification or semantic segmentation of common classes, which mainly leverages its powerful recognition ability via image-text matching, such models fall short on compositional or fine-grained concepts, as introduced in Section 2. CLIP-based VLMs struggle with number-related tasks.

As stated in Section 1, there are insufficient captions for exact number descriptions with paired images from pre-trained web-collected dataset. Paiss et al. (2023) consider object counting task and hypothesize that for current VLMs, pre-trained captions containing an accurate number would become rare when the number of objects exceeds six. By contrast, phrases describing a general form of quantity, e.g., "many", "few" or "a plenty of", would appear more frequently in the caption corpora. From another perspective in numerical cognition, there are two number representation systems in human cognition: an accurate and confident system to discriminate small numerosities (i.e., 1-4 items), referred to as subitizing and mainly based on visual information, and an approximate system to represent larger quantities based on intuitive reasoning, known as number sense (Kajic & Nematzadeh, 2022; Lipton & Spelke, 2003; Kaufman et al., 1949). Such observations support that it is not reasonable to directly treat numbers as class tokens and simply regard the ordinal regression problem as an image-text matching task.

One intuitive idea to solve the insufficient training captions of numbers is to manually construct supplementary training pairs, but it will inevitably cause fast-growing training cost. Alternatively, we cast number indexes/groups into common semantic labels based on the linguistic knowledge from specific tasks, which could be queried by LLMs like ChatGPT, instead of rigidly learning image-to-number alignment from scratch, as shown in Figure 1. The reasons for mapping numbers into linguistic semantic labels are two-folds: one is the potential large quantity of numerical ranges, leading to the redundancy and infeasibility of treating each number as a class token for training; the other is the free "take-away" from pre-trained VLMs. Taking age estimation as an example, we can use "older adult" or "teenager" to describe a person instead of specific ages. It is reasonable to assume that such number-related quantifiable descriptions appear more frequently in the pre-trained captions, and thus are desired to make stronger responses than single numbers (see results in Table 6). The number-to-language mapped semantic labels are served as intermediate classification classes at a coarse level, which is in response to the first aspect of coarse-to-fine approach in Figure 1.

It should be mentioned that the label transformation in this stage is relatively "loose" and holds no restriction on the high accuracy of divided ranges and mapped image pairs, since the exact connection between each number with corresponding textual description depends on the ground-truth labels from a specific task and the language semantics between LLMs and CLIP are not necessarily the same. Nevertheless, it is already enough to generate coarse linguistic semantics and initialize the number-to-language prompts, and thus the pre-trained "numerical knowledge" of CLIP could be elegantly utilized to assist ordinal regression via image-text matching. Inspired by CoOp (context optimization) proposed by Zhou et al. (2022), these initialized prompts will be updated during the training process to iteratively align textual descriptions with corresponding numerical ranges, which are also in consistent with the ground-truth labels of specific tasks. This is consistent to the second aspect of our coarse-to-fine approach in Figure 1, meaning that these intermediate coarse semantic labels could be refined into fine-grained labels in the training process.

## 3.4 Fine-grained Cross-Modal Ranking-based Feature Regularization

The transition from numbers to semantic labels addresses the problem of insufficient numerical training captions, while the insensitivity of these fine-grained descriptions in the contrastive learning process (stated in Section 1) will be solved by our proposed fine-grained cross-modal ranking-based feature regularizer to encourage both semantic and ordinal alignment in CLIP's feature space.

Specifically, current vision-language pre-training is typically conducted by cross-modal contrastive learning, e.g., the InfoNCE loss (Oord et al., 2018). Taking the text-to-image contrastive loss as an example, given an image-text pair $(I, T)$ with $T$ being the anchor and $I_P$ being the positive image sample, all other images in a mini-batch will be regarded as negative samples and therefore be pushed away from the anchor. This training objective could be problematic for ordinal regression task since

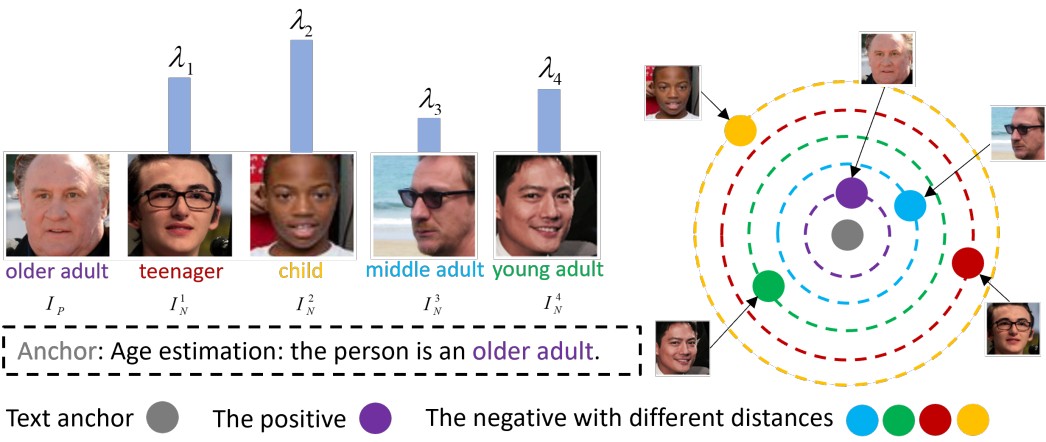

Figure 2: Fine-grained cross-modal ranking-based feature regularization. The cross-modal negative samples are pushed away with ordinal distance alignment.

the inherent ordering relationship among adjacent samples is ignored. As shown in Figure 2, a text can be semantically paired with multiple images with different errors and the subscript $N$ indicates the negative sample. Similar to classification, the prediction error of misclassifying "older adult" as "middle adult" should be lower than that of misclassifying as "teenager" since "middle adult" is closer to "order adult". Therefore, the indiscriminating the relative distances between an image/text anchor with its all negative texts/images will inevitably hinder the learning effect, leading to suboptimal cross-modal representation for ordinal regression.

Mathematically, given a batch of $M$ images $\mathbf{I}_M = \{I_1, I_2, ..., I_M\}$ and paired captions $\mathbf{T}_M = \{T_1, T_2, ..., T_M\}$, CLIP first computes the cosine similarity matrix, denoted by $\mathbf{S} \in \mathbb{R}^{M \times M}$, where each item $S_{i,j}$ indicates the cosine similarity between the $\ell_2$ normalized features of image $i$ and caption $j$. After that, the row- and column-wise cross-entropy losses are computed for standard contrastive learning. Considering the ordering relationship among data, we introduce a weight parameter for each negative sample, which is derived based on cross-modal label distance ranking. The proposed Fine-grained Cross-modal Ranking-based Contrastive loss (FCRC) will keep the negative samples from being improperly minimized and boost a more semantically meaningful representation space aligned with ordinal property. Therefore, the coarse semantic labels initialized by LLMs will be guided to be updated and aligned with image features in an ordinal way, which is consistent to the second aspect of our coarse-to-fine approach in Figure 1.

Formally, with a batch of M semantically aligned image-text pairs $\{(I_i, T_i)\}_{i=1:M}$ and $\ell_2$ normalized embeddings $z^{i=1:M}$ and $w^{i=1:M}$ of each image and text in the batch, the proposed image-to-text FCRC loss is:

$$\mathcal{L}^z_{FCRC} = -\sum_{i=1:M} \frac{1}{M} log \left[ \frac{f\left(z^i, w^i\right)}{f\left(z^i, w^i\right) + \sum_{j \neq i} \lambda^z_{i,j} * f\left(z^i, w^j\right)} \right] \tag{1}$$

where $f\left(z^i, w^j\right) = exp\left(cos\left(z_i, w_j\right)/\tau\right)$ and $\lambda^z_{i,j}$ indicates the contrastive weight of $j_{th}$ negative text sample with respect to $i_{th}$ image anchor in the ordinal training framework. The FCRC loss of text-to-image can be written with a similar format. We adopt an efficient and simplified way to calculate the regularization weight parameter $\lambda_{i,j}$ of negative samples, where the weight parameter should meet two conditions: (1) the weight parameter should be proportional to the relatively ranking distance between the anchor and each negative sample. The larger distance will lead to larger weight penalty to encourage the feature embeddings obeying an ordinal structure. (2) the expectation of $\lambda_{i,j}$ among all negatives of the same anchor is equal to 1 to ensure semantically meaningful feature space. Mathematically, the weight parameter $\lambda_{i,j}$ can be derived as follows:

$$\lambda_{i,j} = Norm(\beta * d_{i,j}) \tag{2}$$

where $\beta$ is a scaling factor, and $d_{i,j}$ is the ranking distance between the anchor and its negative sample.

## 3.5 Overall Objective

Given that RegCLIP is a coarse-to-fine ordinal regression paradigm, the overall training objective is a revised fine-grained cross-modal ranking-based contrastive loss (FCRC) to refine the coarse estimation with the ordinal property constraint, and a regular Euclidean regression loss like MAE to achieve the coarse-to-fine stagewise prediction.

## 4 Experiments

### 4.1 Datasets and Experiment Settings

**Datasets.** We conduct detailed experiments on three different and widely-adopted ordinal regression benchmarks, including age estimation, historical image dating and image aesthetics assessment.

- *Age Estimation*: The task of age estimation is to predict the age of given facial image. The widely used MORPH II (Ricanek & Tesafaye, 2006) dataset is selected to test model performance, which contains 55,134 portraits from 13,618 individuals. Each portrait image is labeled with an age value from 16 to 77. Following popular evaluation protocols (Rothe et al., 2018; Li et al., 2019), only 5,492 images of Caucasian descent are used to remove cross-race interference. 80% of the images are used for training, and others for testing. Overall MAE is reported.

- *Historical Image Dating*: The historical image dating dataset (Palermo et al., 2012) is a benchmark for predicting the decade of given historical colored image. There are five decade categories from the 1930s to 1970s, where each category contains 265 images. Following the general ordinal regression setting (Liu et al., 2018; 2019b), the data of each decade is divided into three parts: 210 for training, 5 for validation, and the rest 50 for testing. Ten-fold cross-validation is adopted. The mean and standard deviation for both classification accuracy and MAE are reported.

- *Image Aesthetics Assessment*: The image aesthetic dataset (Schifanella et al., 2015) contains 13,929 available Flickr photos of four categories, including nature, animal, urban, and people. Each image is judged by at least five different examiners, and five absolute rating scores are used to evaluate the aesthetics quality: "unacceptable", "flawed", "ordinary", "professional", and "exceptional". The ground-truth label of each image is set to be the median among its all gradings. Following Liu et al. (2018), the whole dataset is randomly split into three non-overlapped subsets, 75%, 5% and 20% for training, validation, and testing, respectively. Five-fold cross-validation is used for fair comparison. We report the mean values for both classification accuracy and MAE.

**Experiment Settings.** We adopt the ViT-B/16 image and text encoder of CLIP (Radford et al., 2021) as the model backbone for all experiments. Following OrdinalCLIP (Li et al., 2022), all training data are first resized into $256 \times 256$ and then randomly cropped into $224 \times 224$. We take random horizontal flipping as additional data augmentation. We train the model for 100 epochs with Adam (Kingma, 2014). The text encoder is frozen to keep the pre-trained language semantics intact, only the prompt, image encoder and regressor are trained with a small learning rate of 1e-5. All experiments are conducted on a single NVIDIA 3090 GPU.

### 4.2 Results under fully fine-tuning setting

**Inferring Age from Images.** Table 1 presents the results of existing state-of-the-art methods for ordinal regression. Notable examples include AVDL (Wen et al., 2020), which incorporates an adaptive variance for label distribution within a meta-learning framework, and POE (Li et al., 2021), which considers uncertainty modeling. Upon examining Table 1, it becomes apparent that compared to these specialized approaches, the zero-shot CLIP method only achieves an MAE of 6.09, indicating a limited understanding of numerical or ordinal concept. CoOp (Zhou et al., 2022), which updates the number prompt and achieves an MAE of 2.39, highlights the importance of model fine-tuning. OrdinalCLIP (Li et al., 2022) explicitly models the ranking property of input class embeddings using linear interpolation, resulting in an MAE of 2.32. Notably, RegCLIP achieves a superior MAE of 2.08, surpassing CLIP-based methods and remaining competitive with well-designed state-of-the-art ordinal regression techniques.

**Inferring Decade from Images.** The results presented in Table 2 indicate that zero-shot CLIP still exhibits a poor performance due to its limited understanding of numbers. In contrast, CoOp (Zhou

Table 1: Results of age estimation on MORPH II.

| Methods | MAE | Δ |
|---|---|---|
| AVDL (Wen et al., 2020) | 2.37 | 0.29 |
| DRC-ORID (Lee & Kim, 2020) | 2.26 | 0.18 |
| POE (Li et al., 2021) | 2.35 | 0.27 |
| PML (Deng et al., 2021) | 2.31 | 0.23 |
| MWR (Shin et al., 2022) | 2.13 | 0.05 |
| Zero-shot CLIP (Radford et al., 2021) | 6.09 | 4.01 |
| CoOp (Zhou et al., 2022) | 2.39 | 0.31 |
| OrdinalCLIP (Li et al., 2022) | 2.32 | 0.24 |
| RegCLIP (Ours) | **2.08** | - |

Table 2: Results on Historical Image Dating.

| Methods | Accuracy (%) | MAE |
|---|---|---|
| CNNPOR (Liu et al., 2018) | $50.12 \pm 2.65$ | $0.82 \pm 0.05$ |
| POE (Li et al., 2021) | $54.68 \pm 3.21$ | $0.67 \pm 0.04$ |
| MWR (Shin et al., 2022) | 57.8 | 0.58 |
| GOL (Lee et al., 2022) | 56.2 | 0.55 |
| Ord2Seq (Wang et al., 2023) | 59.5 | 0.53 |
| Zero-shot CLIP (Radford et al., 2021) | $26.08 \pm 0.56$ | $1.48 \pm 0.03$ |
| CoOp (Zhou et al., 2022) | $51.90 \pm 2.60$ | $0.76 \pm 0.06$ |
| OrdinalCLIP (Li et al., 2022) | $56.44 \pm 1.66$ | $0.67 \pm 0.03$ |
| RegCLIP (Ours) | $\mathbf{69.61 \pm 2.02}$ | $\mathbf{0.35 \pm 0.03}$ |

Table 3: Quantitative results on the Image Aesthetic dataset. Both accuracy and MAE are reported.

| Methods | Accuracy(%) - higher is better | | | | | MAE - lower is better | | | | |
|---|---|---|---|---|---|---|---|---|---|---|
| | Nature | Animal | Urban | People | Overall | Nature | Animal | Urban | People | Overall |
| CNNPOR (Liu et al., 2018) | 71.86 | 69.32 | 69.09 | 69.94 | 70.05 | 0.294 | 0.322 | 0.325 | 0.321 | 0.316 |
| SORD (Diaz & Marathe, 2019) | 73.59 | 70.29 | 73.25 | 70.59 | 72.03 | 0.271 | 0.308 | 0.276 | 0.309 | 0.290 |
| POE (Li et al., 2021) | 73.62 | 71.14 | 72.78 | 72.22 | 72.44 | 0.273 | 0.299 | 0.281 | 0.293 | 0.287 |
| GOL (Lee et al., 2022) | 73.8 | 72.4 | 74.2 | 69.6 | 72.7 | 0.27 | 0.28 | 0.26 | 0.31 | 0.28 |
| Zero-shot CLIP (Radford et al., 2021) | 65.24 | 45.67 | 58.78 | 53.06 | 55.68 | 0.461 | 0.557 | 0.468 | 0.524 | 0.502 |
| CoOp (Zhou et al., 2022) | 72.74 | 71.46 | 72.14 | 69.34 | 71.42 | 0.285 | 0.298 | 0.294 | 0.330 | 0.302 |
| OrdinalCLIP (Li et al., 2022) | 73.65 | **72.85** | 73.20 | 72.50 | 73.05 | 0.273 | **0.279** | 0.277 | 0.291 | 0.280 |
| RegCLIP (Ours) | **75.76** | 71.59 | **76.21** | **74.19** | **74.44** | **0.249** | 0.292 | **0.243** | **0.273** | **0.264** |

et al., 2022) and OrdinalCLIP (Li et al., 2022) significantly improve the overall performance compared to zero-shot CLIP. Notably, RegCLIP achieves the highest accuracy of 69.61% and lowest MAE of 0.35, surpassing all other methods by a significant margin.

**Inferring Aesthetics Grading from Images.** Table 3 presents the results of image aesthetics grading. Zero-shot CLIP performs poorly, struggling to differentiate between ordinal concepts. CoOp (Zhou et al., 2022) and OrdinalCLIP (Li et al., 2022) exhibit comparable performance to previous best-performing methods. Unsurprising, RegCLIP outperforms all other methods, achieving an impressive overall accuracy of 74.44% and an MAE of 0.264. The results in individual categories also demonstrate satisfactory performance, highlighting the effectiveness of our method.

### 4.3 RESULTS UNDER FEW-SHOT AND DISTRIBUTION-SHIFT SETTINGS

**Few-shot Learning.** Following OrdinalCLIP (Li et al., 2022), few-shot setting is conducted on the MORPH II (Ricanek & Tesafaye, 2006) dataset to further validate the generalization performance of our method. The training/testing composition is the same as that of OrdinalCLIP (Li et al., 2022) for a fair comparison. The results are presented in Table 4. It is evident that RegCLIP consistently outperforms other methods by a significant margin, particularly in the 1-shot and 2-shot settings. By formulating the problem as a coarse-to-fine paradigm with consideration of incorporating language priors and aligning semantic features in an orderly manner, RegCLIP achieves an impressive MAE of 3.61, compared to 5.09 of CoOp (Zhou et al., 2022) and 4.94 of OrdinalCLIP (Li et al., 2022) under the 1-shot setting. Similar performance gains can be observed across other shot settings, highlighting the effectiveness of RegCLIP. This impressive performance demonstrates the label efficiency of RegCLIP, which is crucial for scenarios with limited training data.

Table 4: The MAE results under few-shot setting on MORPH II.

| # Shots | 1 | 2 | 4 | 8 | 16 | 32 | 64 |
|---|---|---|---|---|---|---|---|
| CoOp (Zhou et al., 2022) | 5.09 | 4.50 | 3.81 | 3.57 | 3.23 | 2.87 | 2.61 |
| OrdinalCLIP (Li et al., 2022) | 4.94 | 4.36 | 3.55 | 3.31 | 3.07 | 2.76 | 2.57 |
| RegCLIP (Ours) | **3.61** | **3.17** | **3.20** | **2.96** | **2.79** | **2.60** | **2.38** |

**Distribution Shift.** Following OrdinalCLIP (Li et al., 2022), data distribution shift experiment is also conducted on the MORPH II (Ricanek & Tesafaye, 2006) dataset to test the model's generalization

Table 5: The MAE results under distribution shift setting on MORPH II. "Num-Cls" denotes the number of discarded classes, and "Per-Dis" means the percentage of discarded samples for each class.

| Num-Cls  Per-Dis | 10-80 | 10-90 | 20-80 | 20-90 | 30-80 | 30-90 | 40-80 | 40-90 |
|---|---|---|---|---|---|---|---|---|
| CoOp (Zhou et al., 2022) | 2.71 | 2.85 | 2.98 | 3.51 | 3.06 | 3.36 | 2.99 | 3.30 |
| OrdinalCLIP (Li et al., 2022) | 2.61 | 2.67 | 2.77 | 3.06 | 2.86 | 3.21 | 2.84 | 3.12 |
| RegCLIP (Ours) | **2.43** | **2.50** | **2.66** | **2.92** | **2.78** | **3.00** | **2.75** | **2.97** |

performance. For the training set, we first randomly select several classes and then discard 80% or 90% of samples in these classes. The rest of the training set is used for training, and the entire test set is used for validation. As can be observed from Table 5, by initializing and updating language prompts, CoOp (Zhou et al., 2022) does not show a severe performance drop, indicating the effectiveness of language information. OrdinalCLIP (Li et al., 2022) further keeps model performance by assuming the linear ranking property of input class word embeddings. RegCLIP obtains the best performance, illustrating that our model can better resist the distribution shift with the assistance of explicit language semantics knowledge and ordered representation constraint. Taking the most severe distribution shift setting (with 40 classes selected and 90% discarded samples) as an example, CoOp(Zhou et al., 2022) and OrdinalCLIP (Li et al., 2022) obtains an MAE of 3.30 and 3.12, respectively. RegCLIP exceeds both with an MAE of 2.97, exhibiting the satisfactory performance.

Table 6: Ablation study of RegCLIP on MORPH II under 1-shot setting.

| Ablation Study | Baseline* | (a) | (b) | (c) |
|---|---|---|---|---|
| Coarse Semantic Label Generation Via LLMs | ✗ | ✗ | ✓ | ✓ |
| Fine-grained Cross-modal Ranking Regularizer | ✗ | ✓ | ✗ | ✓ |
| MAE($\downarrow$) | 4.46 | 4.23 | 3.87 | **3.61** |

**Ablation Study.** The success of RegCLIP is mainly contributed to three important components, namely coarse-to-fine ordinal regression paradigm, coarse semantic label generation via LLMs and fine-grained cross-modal ranking-based regularizer. Since the proposed coarse-to-fine paradigm is fundamental yet distinct with previous state-of-the-art methods with end-to-end classification, we select it as our baseline to validate the effectiveness of the other two components. It is worth mentioning that even without any additional modules, our baseline obtains an MAE of 4.46, which is already higher than 4.94 of OrdinalCLIP (Li et al., 2022) listed in Table 4, under 1-shot setting. This impressive result indicates the effectiveness of the coarse-to-fine paradigm to reduce the learning difficulty, especially for tasks with multiple classes, such as age estimation. Referring back to Table 6, we can see that both the coarse semantic label generation via LLMs, and fine-grained cross-modal ranking regularization could further improve model performance and the joint combination of these modules makes the best performance.

## 5 DISCUSSIONS AND CONCLUSIONS

In this paper, we have presented RegCLIP, a label-efficient coarse-to-fine paradigm for ordinal regression to extend CLIP's potential in a new scenario. We first point out two major reasons for the limited performance of current CLIP-based ordinal regression methods, namely the insufficient numerical training captions and ineffective training objective. To address these issues, we adopt a coarse-to-fine paradigm to reduce the learning difficulty on top of specially designed modules, which is achieved by performing intermediate predictions first and then refining the predictions. Coarse semantic labels generated by LLMs are served as the intermediate labels for coarse prediction. Fine-grained cross-modal ranking-based feature regularization is proposed to subsequently refine the coarse semantic labels with the guidance of ordinal property, through which both semantic and ordinal alignment are achieved in CLIP's feature space. Extensive experimental results show that RegCLIP obtains competitive performance in general ordinal regression tasks, with 10% overall accuracy improvement on historical image dating, 1.74% overall accuracy improvement on image aesthetics assessment, and 1.33 MAE reduction on age estimation under 1-shot setting.

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
