# OpenReview forum: "RegCLIP: A Label-Efficient Coarse-to-Fine Learner for Ordinal Regression"
_ICLR.cc/2024/Conference — ICLR 2024 Conference Withdrawn Submission_

### Official Review · Reviewer_BLNi · 2023-10-24

**Soundness:** 3 good
**Presentation:** 2 fair
**Contribution:** 2 fair
**Rating:** 3
**Confidence:** 4

**Summary:**

This work introduces a novel CLIP-based method for ordinal regression. Two contributions have been made for yielding remarkable performance in multiple datasets. One is to convert numeric labels into semantic labels via CHATGPT. The other one is to develop a fine-grained cross-modal ranking-based loss. In the experiments, a 10% accuracy improvement in historical image dating and a 1.74% improvement in image aesthetics assessment are observed.

**Strengths:**

The literature on the ordinal regression problem is well described, which guides readers to follow the progress made by previous work.

 Casting the regular classification-based and regression-based ordinal regression into cross-modal contrastive learning is a good exploration. Considering the relations between ordinal labels, a fine-grained cross-modal ranking-based loss is used to remedy the issue caused by classical contrastive learning. Ablation studies well verify the effectiveness of the proposed loss.

**Weaknesses:**

The usage of CHATGPT for label transformation is overclaimed. We do not need to use CHATGPT to get the ``older adult`` or ``child`` labels. One can easily imagine some phrases for age description, which may be better than CHATGPT. Also, converting numeric labels into semantic labels is not enough to be one of the main contributions.

The experiments with a few ablations are insufficient. More deep analysis, for example, the sampling in a batch, should be provided to make this submission more solid regarding experiments.

**Questions:**

When reading the last paragraph of the introduction about ``in the coarse stage, we perform intermediate classification using a limited set of classes. This allows for initial decision-making at a coarse level. Subsequently, the fine stage refines the decision within the coarse group assigned by the previous stage``. I cannot well follow the meaning and do not feel confused until reading the method section. It will be better to rewrite the sentences and make them clear.

``one is the potential large quantity of numerical ranges, leading to the redundancy and infeasibility of treating each number as a class token for training.`` What does the redundancy mean here?

For the training, the paper mentions a coarse-to-fine diagram. How many stages are there for the training?

---

### Official Review · Reviewer_oR5C · 2023-10-30

**Soundness:** 2 fair
**Presentation:** 3 good
**Contribution:** 2 fair
**Rating:** 5
**Confidence:** 5

**Summary:**

In this paper, the authors propose a novel coarse-to-fine learning approach for ordinal regression. The first stage of their method leverages Language and Vision Models (LLMs) to generate descriptive labels instead of numerical ranks in a label-efficient manner. These labels are then used for intermediate text-image classification through prompt tuning. In the second stage, the authors fine-tune the image decoder using a new rank-based contrastive loss. Extensive experiments demonstrate that their approach, named RegCLIP, outperforms prior state-of-the-art methods on three widely recognized benchmarks.

**Strengths:**

Simplicity:
-	Combines ordinal regression with LLMs and utilizes LLMs to enhance label efficiency.
-	Design a rank-based contrastive loss to ensure alignment of ordering and semantics in the CLIP feature space.
-	Propose a staged process to further enhance performance.
Empirical Evaluation:
Extensive experiments validate the effectiveness of our proposed method in classifying ordinal tasks. Additionally, these experiments demonstrate its potential in few-shot learning and addressing data distribution shifts.

Writing:
The paper is well-organized and written in a clear manner. The main results seems to be  easily reproducible.

**Weaknesses:**

1.	Major concerns:
a) It appears that LLMs have a minimal contribution to the ordinal task. Obtaining general descriptions for different age ranges seems achievable without the use of LLMs. Additionally, it is unclear how LLMs can generate semantic labels for the historical image dating task.
b) Theory validation is necessary. The proposed FCRC introduces an additional distance-weighting term compared to the normal contrastive loss. It may not be a novel contribution given the current research landscape. Providing more mathematical deduction could aid in better understanding.
c) Visualization results could be beneficial in demonstrating the superiority of RegCLIP in terms of ordinality.
d) A more comprehensive analysis is needed to compare RegCLIP with previous methods, such as L2RCLIP [1].
2.	Minor concerns:
a) It is expected to conduct experiments on more datasets and settings, including MORPH (Setting B, C, D), CLAP2015, CACD, etc.
b) If there exists another architecture for the regressor, please provide additional details.
[1] Learning-to-Rank Meets Language: Boosting Language-Driven Ordering Alignment for Ordinal Classification. NeurIPS, 2023.

**Questions:**

Please refer to the Weaknesses.

---

### Official Review · Reviewer_i2rY · 2023-10-30

**Soundness:** 3 good
**Presentation:** 2 fair
**Contribution:** 2 fair
**Rating:** 5
**Confidence:** 4

**Summary:**

This paper focuses on the ordinal regression task and proposes an CLIP-based coarse-to-fine method named RegCLIP. Firstly, this paper generates coarse labels via large language models. Secondly, a fine-grained cross-modal ranking-based loss is designed to further improve the precision of predictions. The proposed method achieves superior performance than state-of-the-art methods on three public datasets.

**Strengths:**

1. Based on the fact that learning from a staged process is often more effective and easier than directly learning from multiple precise values, this paper proposes a coarse-to-fine ordinal regression paradigm.
2. In the coarse stage, the authors adopt large language models to generate coarse semantic labels, which is label-efficient.In the fine stage, the authors propose a novel fine-grained cross-modal ranking-based feature regularization loss to further improve the classification performance.
3. Extensive experiments show that the proposed method is superior to state-of-the-art methods and each component is effective.

**Weaknesses:**

1. The contribution that generating coarse semantic labels via large language models is weak. According to the description, the coarse semantic labels could be obtained by simply inputting a task-related prompt to LLMs.
2. The description of how to gradually update prompts in section 3.3 is missing, which is important to the coarse-to-fine process.
3. These three datasets are all small, and a small number of samples being misclassified can have a significant impact on the results.

**Questions:**

What is the input for the regressor? Coarse estimation requires a specific explanation.

---

### Official Review · Reviewer_DJCj · 2023-11-01

**Soundness:** 3 good
**Presentation:** 3 good
**Contribution:** 3 good
**Rating:** 8
**Confidence:** 5

**Summary:**

In this paper, CLIP-based ordinal regression method is proposed. First, it utilizes the LLM to group the ordinal classes into a few classes. Then, the CLIP image encoder and regressor are trained by using weighted contrastive loss between language features and image features. Here, the weights are computed based on the difference of coarse LLM generated classes. The proposed algorithm outperforms conventional methods on various datasets for age estimation, historical image classification, and aesthetic quality assessment.

**Strengths:**

- The paper is easy to follow and well organized.
- The proposed algorithm is simple but effective way to utilize CLIP for ordinal regression tasks.
- The proposed algorithm shows good results on various datasets.

**Weaknesses:**

1. Even though the authors mentioned that they will release the codes, I think there are some missing details which are important for reproducibility and further researches.

     - Section 3.5: Exactly, what is the revised FCRC loss?
     - How the fine-grained labels are predicted?
     - What is batch size M?
     - In Eq(2), what is the default value for $\beta$?
     - In Eq(2), how the scaled ranking differences are normalized?
     - What are language templates for historical image classification and aesthetic quality assessments?


2. It would be good to have comparison of feature space visualizations for ablated methods. I think it can provide insights and intuitive understanding of each design choice for readers.


3. MORPH II dataset has many different evaluation protocols. However, the results for only one setting are provided in the paper. So, it may be more useful to have more comparisons on MORPH II dataset.


4. There are some other popular datasets for age estimation (e.g. CACD, UTK, CLAP2015, Adience, …). So, it would be better if the results on those datasets are compared in the paper.

**Questions:**

The proposed algorithm is simple but makes sense for me. However, it omits some important details in the paper and does not contain many analysis of the proposed algorithm, as I mentioned in the weakness section. (Please see the weakness section for other concerns.) Nevertheless, in overall, I'm positive about the acceptance of this paper, because it outperforms all conventional methods with meaningful performance gaps and also authors described that the source codes will be released for algorithm details.

---

### Official Review · Reviewer_YX7j · 2023-11-02

**Soundness:** 2 fair
**Presentation:** 3 good
**Contribution:** 2 fair
**Rating:** 3
**Confidence:** 5

**Summary:**

This paper proposes a novel ordinal regression framework named RegCLIP. The authors employ language prior information to refine the prediction results. Specifically, the authors first generate intermediate classes for different downstream tasks by using LLMs and then refine them into fine-grained labels. Meanwhile, the authors propose a novel fine-grained cross-modal ranking loss to further improve the ordinal alignment performance. Extensive experiments on publicly available datasets validate the effectiveness of the proposed approach.

**Strengths:**

This paper has the following strengths:
1. The graphical representation of this paper is clear;
2. The framework proposed in this paper is relatively novel, utilizing LLM to provide a priori knowledge for downstream tasks and for inter-modal alignment training;
3. Excellent performance of the experimental results.

**Weaknesses:**

This paper has the following weaknesses:
1.	Although the idea of the paper is novel, the contribution is very limited. The approach of using language prior knowledge provided by LLM is too naïve and fails to illustrate the superiority of LLM in providing prior information.
2.	Some necessary explanations are missing. For example, the paper points out that previous work ignored numerical knowledge in cross-modal feature space, but does not explain why the previous framework ignored the knowledge, and also does not point out why the proposed approaches can capture this type of knowledge.
3.	Some of the symbols in the paper lack the necessary comments.
4.	The paper is not well written. The overall layout could be improved.

**Questions:**

The reviewers' main questions focus on the significance of LLM in the proposed framework and technological contributions.
1. The usage of LLMs to provide prior knowledge or fine-tune datasets for downstream tasks has been widely used in the community. However, this paper only uses a very simple prompt to call ChatGPT to generate coarse semantic labels, and the paper does not indicate the advantages of this approach.
2. The current way of generating coarse semantic labels is weakly correlated to specific inputs, currently appearing to be task-related only, and the selection is very subjective without quantitative metrics.
3. In terms of the contribution of LLMs to downstream tasks, the currently proposed frameworks contribute weakly. I suggest that the authors could consider some VLM training approaches (e.g., LLaVA) to generate command-follow datasets for specific downstream tasks.
4. Experimental implementation details are missing. The choice of hyperparameters in the method is not defined.